

# Physical activity questionnaire for older children (PAQ-C): Arabic translation, cross-cultural adaptation, and psychometric validation in school-aged children in Saudi Arabia

Mohamed Sherif Sirajudeen[1], Mohamed Waly[2], Md. Dilshad Manzar[3], Mazen Alqahtani[1], Msaad Alzhrani[1], Ahmad Alanazi[1], Radhakrishnan Unnikrishnan[1], Hariraja Muthusamy[1], Rashmi Saibannavar[1] and Wafa Alrubaia[1]

[1] Department of Physical Therapy and Health Rehabilitation, College of Applied Medical Sciences, Majmaah University, Majmaah, Saudi Arabia
[2] Department of Medical Equipment Technology, College of Applied Medical Sciences, Majmaah University, Majmaah, Saudi Arabia
[3] Department of Nursing, College of Applied Medical Sciences, Majmaah University, Majmaah, Saudi Arabia

## ABSTRACT

The validity of the Physical Activity Questionnaire for Older Children (PAQ-C) has been mostly studied in North America and Europe. We investigated the psychometric validation of the Arabic version of the PAQ-C in students in Saudi Arabia. The students ($n$ = 327, age = 8–14 years) of six primary schools in the Majmaah region participated in the study. Participants completed the PAQ-C, and their demographics were recorded. The PAQ-C scores satisfied the following factor analysis assumptions: diagonal elements of the anti-image correlation matrix (>0.5), Bartlett's test of sphericity ($p$ <0.001), determinant (>0.00001), Kaiser–Meyer–Olkin test of sampling adequacy (>0.8), and communality (all values > 0.2). Exploratory factor analysis results were inconclusive, with two measures favoring a 2-factor solution (Kaiser's criteria (Eigenvalue ≥ 1), and cumulative variance rule (>40%)); whereas, the scree test and the Monte Carlo parallel analysis favored a 1-factor structure. The confirmatory factor analysis favored a 1-factor solution: highest CFI, lowest RMSEA, non-significant $\chi^2$ statistics, and lowest $\chi^2$/df. The values of item-total correlation, corrected item-total correlation, and Cronbach's alpha if an item was deleted, ranged from 0.20–0.57, 0.42–0.64, and 0.70–0.75, respectively. The PAQ-C showed a Cronbach's alpha of 0.74. A 1-factor structure of the Arabic version of the PAQ-C had adequate psychometric validity in schoolchildren in Saudi Arabia.

## INTRODUCTION

Physical inactivity is a significant risk factor for cardiovascular disorders and a broad spectrum of chronic diseases comprising obesity, diabetes mellitus, hypertension,

Corresponding author
Md. Dilshad Manzar,
m.manzar@mu.edu.sa

osteoarthritis, osteoporosis, and depression (*Warburton, Nicol & Bredin, 2006*). As per a report issued by the World Health Organization (WHO), 60% of the global population is physically inactive; thus, a sedentary lifestyle emerges as a menace to global health (*World Health Organization, 2003*). A recent systematic review summarizing 18 articles published between 2007 and 2017, found that physical inactivity was highly prevalent among Saudi adolescents (*Alasqah et al., 2021*). Another systematic review summarized that the majority of children and youth in Saudi Arabia did not meet the criteria for moderate to vigorous physical activity. Physical inactivity is more prevalent in Saudi females, and the problem starts in the early school years (*Al-Hazzaa, 2018*).

Physical inactivity during childhood considerably impacts the level of physical activity and obesity in adulthood, which sequentially relates to the incidence of lifestyle-related disorders occurring in adulthood (*Isa et al., 2019*; *Dietz, 1998*; *Field, Cook & Gillman, 2005*; *Telama et al., 2005*). Regular physical activity in childhood promotes optimal body mass index (BMI), cardiopulmonary fitness, and psychosocial wellbeing (*Denton et al., 2013*; *Tomson et al., 2003*; *Suter & Hawes, 1993*). A valid and reliable assessment of physical activity is imperative to investigate the efficacy of measures, enhance physical activity, and track the related health benefits (*Platat & Jarrar, 2012*). Physical activity can be determined both by objective means, such as indirect calorimetry and accelerometry-based assessments, or subjectively, by using a self-report questionnaire. Indirect calorimetry delivers a precise measure of physical activity, but due to its invasive nature, it is unfeasible for use in field-based studies involving larger populations (*Wang et al., 2016*).

Physical activity assessments using accelerometers provide accurate measures; however, they are uneconomical and often result in adherence concerns among children, such as feeling awkward or not remembering to wear the device (*Corder, Brage & Ekelund, 2007*). On the other hand, subjective measures, for example, self-reported questionnaires, are inexpensive, easy to administer, and enable the collection of contextual details of physical activities that cannot be obtained through objective measures. Hence, self-reported questionnaires emerge as a universally preferred method to assess the physical activity of children in research involving larger populations (*Matthews & Welk, 2002*).

Intentions may reveal motivation, and help determine social actions (*Terry & O'Leary, 1995*). Constructs based on the self-determination theory (SDT) have been used to measure motivation to perform physical activity in various age groups (*Sebire et al., 2013*). Different dimensions of motivation as envisaged in the SDT such as personal, psycho-social, and environmental aspects are relevant for physical activity measurements (*Plotnikoff et al., 2013*). Various aspects of autonomous intention including intentions to exercise, and consistency between intentions to exercise are associated with actual exercising during leisure time (*Chatzisarantis, Biddle & Meek, 1997*). In summary, dimensions of measures to assess motivation to perform physical activity are determined by SDT theory, many of these autonomous dimensions of motivation are related to physical activity. Children's physical activity is usually irregular and difficult for them to recall in terminologies used in self-reported questionnaires, such as intensity, duration,

and frequency (*Baquet et al., 2007*; *Hussey, Bell & Gormley, 2007*). Hence, questionnaires meant to assess children's physical activity should be constructed to reduce the intellectual, recall, and calculation demands to a justifiable level (*Kremers et al., 2005*). The availability of a valid and reliable subjective assessment method of physical activity remains a challenge in Saudi Arabia. This is due to the lack of availability of culturally adapted physical activity questionnaires in the Arabic language (*Hussey, Bell & Gormley, 2007*). The majority of the physical activity studies involving children in Saudi Arabia either utilized the English questionnaire or translated a general physical activity questionnaire, which is not specific for children (*Tomson et al., 2003*; *Suter & Hawes, 1993*; *Wang et al., 2016*; *Corder, Brage & Ekelund, 2007*). Hence, it is crucial to develop a culturally adaptable physical activity questionnaire for children in the Arabic language.

The physical activity questionnaire for older children (PAQ-C), developed by *Kowalski, Crocker & Donen (2004)*, is a seven-day recall self-administered questionnaire to evaluate moderately to vigorous physical activity in children belonging to the 8–14 years age group. The PAQ-C employs memory cues to facilitate the participant's recollection of their physical activities, which makes the PAQ-C an apt tool to use with children (*Thomas & Upton, 2014*). The psychometric properties of the PAQ-C were investigated among various English populations, including Canadian, British, Euro-American, and African-American children (*Thomas & Upton, 2014*; *Janz et al., 2008*; *Moore et al., 2007*). The results of these studies reported acceptable to good internal consistency, test-retest reliability, sensitivity to distinguish gender-related variations, convergent validity, and construct validity (*Crocker et al., 1997*; *Kowalski, Crocker & Faulkner, 1997*). Additionally, the investigators reported normal scale distribution and group mean-variance demonstrating the instruments' ability to determine various levels of physical activities (*Crocker et al., 1997*; *Kowalski, Crocker & Faulkner, 1997*).

Although there is a significant amount of evidence of the instrument's psychometric properties among English children, research indicates that the PAQ-C may fail to exhibit external validity when administered in other populations with different races or ethnicities. Moreover, the factor construct of the PAQ-C was also reported to differ according to the population studied (*Thomas & Upton, 2014*). The studies among Italian, United States and Turkish children reported a 2-factor model, whereas, the research among Chinese and Czech children determined a 1-factor model (*Wang et al., 2016*; *Thomas & Upton, 2014*; *Gobbi et al., 2016*; *Erdim, Ergün & Kuğuoğlu, 2019*; *Cuberek, Janíková & Dygrtfytfn, 2021*). Hence, it is crucial to consider factors such as ethnicity, race, culture, and language of the target population before the translation of the PAQ-C. We hypothesized that the Arabic version of the PAQ-C will exhibit adequate reliability, internal homogeneity, and favorable item analysis. Additionally, we hypothesized that the factor structure validated previously in the Asian population may be valid for the Arabic version of the PAQ-C in a sample of Saudi school children. Therefore, we performed an Arabic translation, cross-cultural adaptation, and psychometric validation in a sample of school-aged children in Saudi Arabia.

## MATERIALS AND METHODS

### Participants

Male and female school-going children aged 8 to 14 years from the Majmaah region, Kingdom of Saudi Arabia took part in this cross-sectional study. We determined the sample size in this study using the criteria given by MacCallum for factor analysis (*MacCallum et al., 1999*). As the study published by *Wang et al. (2016)* is close concerning ethnicities, therefore, we used the factor loading values published in their article to determine the sample size. We used these factor loading values in the *Wang et al. (2016)* study to calculate communality for all the PAQ-C items using the formula given in Yong, and Pearce's article (*Yong & Pearce, 2013*). The average factor loading was 0.56 implying a good level of correlations between measured variables, *i.e.*, item scores, and the factor score (*Comrey & Lee, 2013*). However, the calculated value of average communality was lower, *i.e.*, 0.36. *Wang et al. (2016)* found a 1-factor solution for the PAQ-C in children in Hong Kong, therefore, the number of determinants was high, *i.e.*, nine items for 1 factor. In summary, evidence from a moderate average factor loading, low average communality, and high over-determination (*i.e.*, number of items per factor), a sample size of a little over 100 would be adequate for a study performing either EFA or CFA only. As, we intended to perform both EFA and CFA, as well as taking into account the usual attrition rate, a sample size of a little over 300 was expected to be appropriate (*MacCallum et al., 1999*; *Manzar, Jahrami & Bahammam, 2021*).

The study population comprised 1,892 children in the target age group belonging to 15 different schools located in the Majmaah region. Out of which, six schools were identified using the cluster sampling method, and students from these schools were selected based on convenient sampling. Most of the students from the selected schools agreed to participate in the study. The sample consisted of 327 students belonging to grades 2 to 6. The parents of the participating students provided signed informed consent. In addition, consent was obtained from the participating students before data collection.

Data collection was performed at their respective schools in the morning hours on their regular school days during February and March 2020. The participants filled the PAQ-C in their respective classrooms in the presence of their class teacher. Members of the research team displayed the items of the PAQ-C using the projector, explained the items, and clarified the doubts of the participants. The children completed the PAQ-C approximately in 20 min. The children with apparent disabilities and those who reported sick or any event that prevented them from performing normal physical activities in the preceding 7 days were excluded. The Majmaah University Ethics committee provided the ethical approval for this study (MUREC-Feb.19/COM-2020/21-5). All methods of this study were carried out in accordance with the principles of the declaration of Helsinki.

### Procedure

Details related to personal characteristics, for example, age, gender, hand dominance, height, weight, and BMI were gathered. The hand reportedly used by the participant to perform daily activities like writing, eating, and throwing a ball was determined as

dominant (*Omar et al., 2018*). Height and weight were determined in bare feet by using Solo Eye-level Physician Scale (Detecto Inc., Webb City, MO, USA) and tabulated to the nearest 1 cm and 100 g, respectively. BMI was measured using standard norms by dividing the weight of the participant by his/her height in squared meters (*Gerver & de Bruin, 2001*).

## Physical activity questionnaire-children (PAQ-C)

The physical activity of the participants was measured using the Arabic version of the PAQ-C, comprising ten items, nine of which were scored on a five-point rating scale where higher scores indicate a higher level of activity. The first item of the PAQ-C consists of 22 common sports and leisure activities for which the participants select the score based on the frequency of the activity performed during the preceding seven days on a five-point rating scale (1 = no activity at all, 2 = 1–2 times, 3 = 3–4 times, 4 = 5–6 times and 5 = 7 times or more), after which a mean composite score was calculated (*Kowalski, Crocker & Donen, 2004*). The remaining eight items address physical activities performed during the day, for example, physical education classes, recess time, lunchtime, as well as afterschool activities on weekday evenings and weekends, and finally a summary for all days of the week. The mean score of the first nine items is the summary score of the PAQ-C. The tenth item inquires about any unusual circumstances (*e.g.*, sickness) that affected/prevented the child's physical activity in the seven days preceding the assessment (*Kowalski, Crocker & Donen, 2004*).

## Physical activity questionnaire-children (PAQ-C): Arabic translation and cross-cultural adaptation

Two bilingual translators fluent in both English and Arabic translated the original PAQ-C from English to Arabic. Among the two bilingual translators, the first translator was a Physical therapist with a Ph.D. qualification from the USA who was aware of the construct area of the tool and was exposed to both cultural backgrounds. The second was an officially certified bilingual translator and aware of native terminologies. Later, two bilingual translators independent of the earlier translators with a similar background performed the back translation of the Arabic document into English based on the "translation and back-translation method" (*Sousa & Rojjanasrirat, 2011*). The original English version of the PAQ-C and the back-translated version of the PAQ-C were similar without any changes in meaning.

An expert panel consisting of 13 members which included three physical therapists, three exercise and sports science experts, three pediatricians, three physical education experts, and one public health expert evaluated for comprehensibility and performed the cultural adaptation of the PAQ-C to suit the Arabic population. Among the activities listed in item 1, the activities that are not typically performed in Arabian countries (*e.g.*, rowing, American football, street hockey, floor hockey, cross-country skiing, and ice hockey) were deleted based on the opinions of the expert panel. Additionally, physical activities such as in-line skating, baseball, soccer, and ice skating were substituted with roller skating, tennis, football, karate, and other martial arts, respectively. As a result of

these changes, item 1 of the Arabic version of PAQ-C consists of a list of 16 physical activities instead of 22 activities, as in the original PAQ-C. A sample of 15 school children between the ages of 8 and 14 participated in the pilot study to check the comprehensibility of the Arabic version of the PAQ-C. The bilingual translators, expert panel members, and participants of the pilot study agreed that the words used in the translated version of the PAQ-C were simple and easy to understand for the children of the target age group.

## Statistical analyses

All statistical analyses were performed using SPSS version 26.0 and a syntax program (parallel.sps) developed by O'Connor (*O'Connor, 2000*). Participants' characteristics and preliminary item analysis were assessed by measures of descriptive statistics and distribution indices. Internal homogeneity and reliability were determined by Spearman's correlation test and Cronbach's alpha. Factor analysis was performed on two sub-samples. The two sub-samples were split randomly using the coin method (heads and tails). Principal axis factoring was selected in the exploratory factor analysis (EFA) because, in the EFA sub-sample ($n = 190$), some items were not normally distributed as determined by the criteria of the absolute z-score value of skewness and kurtosis (3.29) (*Kim, 2013*). The Promax rotation method was used because factors were expected to be correlated (*Erdim, Ergün & Kuğuoğlu, 2019*). A confirmatory factor analysis (CFA) was done on a second sub-sample (randomly split) using maximum likelihood extraction. Bollen-Stine bootstrapping was employed to manage deviations from multivariate normality. Inter-factor correlations (IFC), standardized factor loadings, and multiple fit indices from different categories were estimated. A comparative CFA was performed on the factor structures shown by the EFA and those reported in the previous studies (*Gobbi et al., 2016*; *Erdim, Ergün & Kuğuoğlu, 2019*). A value of 0.90 and above for the comparative fit index (CFI) and goodness of fit index (GFI) indicated adequate validity (*Hu & Bentler, 1999*; *Brewer & Jones, 2002*; *Butragueño & Peinado, 2014*; *Simón, Fernandez & Contreras, 2017*; *Otero-Saborido, González-Jurado & Lluch, 2012*; *Manzar et al., 2018*). A value of 0.08 or less for the root mean square residual (RMR) and root mean square error of approximation (RMSEA) implied a good fit (*Hu & Bentler, 1999*; *Manzar et al., 2018*). A non-significant chi-square test and $\chi^2/df$ of less than 3 indicated a good fit as well (*Hu & Bentler, 1999*; *Manzar et al., 2018*).

## RESULTS

### Participants' characteristics

Summary statistics of the participating Saudi schoolchildren are shown in Table 1. Approximately half of the participants were in the 10–12 years age group. The majority of children were girls (52.3%). Most of the participating children (67.6%) were enrolled in grades 4–5. The average value of the BMI was 19.40 ± 4.84 kg/m$^2$. Most of the participating children (94.8%) reported right-hand dominance.

**Table 1 Participant characteristics.**

| Characteristics | Mean ± SD/frequency |
|---|---|
| Age (yr) | 10.7 ± 1.3 (8–14) |
| 8–10 | 68 (20.8) |
| 10–12 | 163 (49.8) |
| 12–14 | 96 (29.4) |
| Gender | |
| Boy | 156 (47.7) |
| Girl | 171 (52.3) |
| Grade | |
| 2 | 16 (4.9) |
| 3 | 43 (13.1) |
| 4 | 47 (14.4) |
| 5 | 131 (40.1) |
| 6 | 90 (27.5) |
| BMI (kg/m$^2$) | 19.40 ± 4.84 |
| Hand dominance | |
| Left | 17 (5.2) |
| Right | 310 (94.8) |

Note:
SD, standard deviation; BMI, body mass index.

**Table 2 Sample size adequacy measures of the physical activity questionnaire-children (PAQ-C).**

| Measures | Values | | |
|---|---|---|---|
| | Total sample ($N$ = 327) | CFA sub-sample ($n$ = 137) | EFA sub-sample ($n$ = 190) |
| Anti-image matrix | 0.75–0.89 | 0.65–0.88 | 0.73–0.89 |
| Bartlett's test of Sphericity | $\chi^2$ (36) = 566.3, $p$ < 0.001 | $\chi^2$ (36) = 281.6, $p$ < 0.001 | $\chi^2$ (36) = 318.28, $p$ < 0.001 |
| Determinant | 0.172 | 0.12 | 0.18 |
| Kaiser-Meyer-Olkin Test of Sampling Adequacy (KMO) | 0.84 | 0.82 | 0.83 |
| Communality | 0.30–0.58 | 0.22–0.59 | 0.37–0.59 |

Note:
EFA, exploratory factor analysis; CFA, confirmatory factor analysis.

## Factor analysis

### Sample adequacy measures

As the factor analysis was performed after splitting the sample into two sub-samples for carrying out EFA and CFA separately, the suitability of the PAQ-C score distribution for the factor analysis was determined by assessing the diagonal element of the anti-image correlation matrix, Bartlett's test of sphericity, determinant, the Kaiser-Meyer-Olkin (KMO) test of sampling adequacy, communality, and inter-item correlation coefficients (see Tables 2–3) in both the sub-samples (*Manzar et al., 2018*; *Field, 2013*; *Child, 2006*). The PAQ-C scores in both sub-samples fulfilled the criteria of all six measures, namely,

Table 3 Inter-item correlation matrix of the physical activity questionnaire-children (PAQ-C).

| Items of the PAQ-C | PAQ-C-1 | PAQ-C-2 | PAQ-C-3 | PAQ-C-4 | PAQ-C-5 | PAQ-C-6 | PAQ-C-7 | PAQ-C-8 | PAQ-C-9 | |
|---|---|---|---|---|---|---|---|---|---|---|
| PAQ-C-1 | | 0.06 | 0.27** | 0.09 | 0.44** | 0.43** | 0.40** | 0.34** | 0.41** | CFA sub-sample |
| PAQ-C-2 | | | 0.18* | −0.01 | 0.10 | 0.14 | 0.10 | 0.20* | 0.19* | |
| PAQ-C-3 | | | | 0.04 | 0.18* | 0.20* | 0.32** | 0.16 | 0.19* | |
| PAQ-C-4 | | | | | 0.11 | 0.18* | 0.23** | 0.08 | 0.22* | |
| PAQ-C-5 | | | | | | 0.56** | 0.40** | 0.39** | 0.50** | |
| PAQ-C-6 | | | | | | | 0.37** | 0.33** | 0.44** | |
| PAQ-C-7 | | | | | | | | 0.35** | 0.34** | |
| PAQ-C-8 | | | | | | | | | 0.52** | |
| PAQ-C-9 | | | | | | | | | | |
| PAQ-C-1 | | 0.12 | 0.21** | 0.14* | 0.32** | 0.28** | 0.27** | 0.29** | 0.28** | EFA sub-sample |
| PAQ-C-2 | | | 0.25** | 0.08 | 0.15* | 0.04 | 0.14 | 0.15* | 0.26** | |
| PAQ-C-3 | | | | 0.31** | 0.14 | 0.22** | 0.32** | 0.25** | 0.27** | |
| PAQ-C-4 | | | | | 0.16* | 0.22** | 0.34** | 0.19** | 0.29** | |
| PAQ-C-5 | | | | | | 0.35** | 0.38** | 0.40** | 0.35** | |
| PAQ-C-6 | | | | | | | 0.23** | 0.29** | 0.40** | |
| PAQ-C-7 | | | | | | | | 0.41** | 0.28** | |
| PAQ-C-8 | | | | | | | | | 0.30** | |
| PAQ-C-9 | | | | | | | | | | |

Notes:
\* $p < 0.05$.
\*\* $p < 0.01$.
EFA, exploratory factor analysis; CFA, confirmatory factor analysis.

all the diagonal elements in the anti-image correlation matrix were more than 0.5 (in fact, the least observed value was 0.65) (*Manzar et al., 2018*; *Child, 2006*); the significant values of Bartlett's test of sphericity implied that there was a linear relationship between PAQ-C items, and the absence of inter-item correlation coefficients with the absolute value r = 1 was also noted, which implies an absence of singularity (*Manzar et al., 2018*; *Child, 2006*); there was no major concern of multicollinearity bias, as established by the determinant's value of more than 0.00001 (*Manzar et al., 2018*; *Child, 2006*); and there was a meritorious level of shared variance among the PAQ-C item scores, as evidenced by a KMO value of 0.82 and above for both the sub-samples (see Tables 2–3) (*Manzar et al., 2018*; *Field, 2013*; *Child, 2006*). Finally, communality values were also more than the minimum recommended value of 0.2 (see Table 2) (*Child, 2006*).

*Exploratory factor analysis*

In EFA, two measures of factor retention, namely, Kaiser's criteria (Eigenvalue ≥ 1), and cumulative variance rule (>40%) found that a 2-factor solution was appropriate (see Table 4). As per Kaiser's criteria, two factors were found to have an Eigenvalue of more than 1 (Table 4). As per cumulative variance criteria, the 2-factor solution showed a combined variance of 47.84% (Table 4). Manual inspection of the point of inflection on the actual Eigenvalue curve with the number of factors showed a 1-factor solution (Table 4, Fig. 1). Moreover, when a more robust measure of factor retention, the Monte Carlo parallel analysis, was used, it indicated a 1-factor structure (see Fig. 1) for the PAQ-C in the

**Table 4 Summary of the factor retention measures used in exploratory factor analysis of the physical activity questionnaire-children (PAQ-C).**

| Number of Factors | Eigenvalue | Cumulative variance explained (%) | Above point of inflection on Scree plot | Decision to extract | | |
|---|---|---|---|---|---|---|
| | | | | Kaiser's criteria (Eigenvalue ≥ 1) | Cumulative variance rule (>40%) | Scree test |
| 1 | 3.21 | 35.69 | Yes | √ | √ | √ |
| 2 | 1.09 | 47.84 | No | √ | √ | ✗ |
| 3 | 0.96 | 58.55 | No | ✗ | ✗ | ✗ |
| 4 | 0.78 | 67.23 | No | ✗ | ✗ | ✗ |
| 5 | 0.72 | 75.26 | No | ✗ | ✗ | ✗ |

Note:
√ indicates extraction criteria fulfilled, ✗ indicates otherwise.

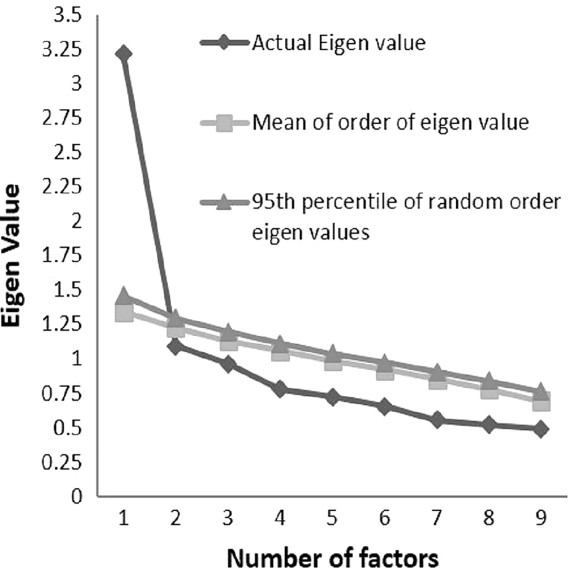

**Figure 1 Parallel analysis sequence plot of the physical activity questionnaire-children (PAQ-C)** Monte Carlo parallel analysis with principal components and random normal data generation.

participating Saudi schoolchildren (*O'Connor, 2000*; *Manzar et al., 2018*; *Brown, 2015*; *Manzar et al., 2016*). The interpretation of the parallel analysis outcome was based on the recent recommendation that suggests retaining the number of factors for which the actual Eigenvalue is greater than the 95[th] percentile of the random order eigenvalues. In this study, for the first factor, the value of the actual Eigenvalue (3.21) was more than the 95[th] percentile of the random ordered Eigenvalue (1.46) in PCA principal component analysis (Table 5, Fig. 1) (*O'Connor, 2000*; *Manzar et al., 2018*; *Brown, 2015*; *Manzar et al., 2016*).

All but one (PAQ-C-2) item of the PAQ-C fulfilled the minimum level of correlation between the latent construct and their loading items, as determined by the criteria given by *Comrey & Lee (1992)* (see Table 6) (*Comrey & Lee, 2013*).

**Table 5 Parallel analysis (Monte Carlo PA) output of the physical activity questionnaire-children (PAQ-C).**

| Number of factors | Actual eigenvalue from PCA | Random order eigenvalues (means) | Random order eigenvalues (95th percentile) |
|---|---|---|---|
| 1* | 3.21* | 1.34 | 1.46* |
| 2 | 1.09 | 1.23 | 1.30 |
| 3 | 0.96 | 1.13 | 1.20 |
| 4 | 0.78 | 1.06 | 1.11 |
| 5 | 0.72 | 0.99 | 1.04 |
| 6 | 0.66 | 0.92 | 0.97 |
| 7 | 0.56 | 0.85 | 0.91 |
| 8 | 0.52 | 0.78 | 0.84 |
| 9 | 0.49 | 0.69 | 0.77 |

Note:
* For first factor, value of the actual Eigenvalue (3.21) was more than the 95[th] percentile of the random ordered Eigenvalue (1.46) in PCA principal component analysis.

**Table 6 Descriptive statistics of the physical activity questionnaire (PAQ-C) scores.**

| Items of the PAQ-C | Factor loadings in EFA | | Item-total Correlation | Corrected Item-total Correlation | Cronbach's alpha if item deleted | Mean ± SD | Skewness | | Kurtosis | |
|---|---|---|---|---|---|---|---|---|---|---|
| | Factor-1 | Factor-2 | | | | | Statistic (SE) | z | Statistic (SE) | z |
| PAQ-C-1 | 0.47 | | 0.47 | 0.52* | 0.72 | 1.76 ± 0.53 | −0.11 (0.14) | −0.82 | −0.87 (0.27) | −3.23 |
| PAQ-C-2 | | 0.28 | 0.20 | 0.42* | 0.75 | 3.70 ± 1.21 | −0.48 (0.14) | −3.58 | −0.60 (0.27) | −2.21 |
| PAQ-C-3 | | 0.70 | 0.36 | 0.57* | 0.73 | 3.02 ± 1.43 | −0.12 (0.14) | −0.86 | −1.24 (0.27) | −4.59 |
| PAQ-C-4 | | 0.53 | 0.33 | 0.48* | 0.73 | 2.27 ± 1.35 | 0.61 (0.14) | 4.55 | −0.93 (0.27) | −3.46 |
| PAQ-C-5 | 0.79 | | 0.49 | 0.64* | 0.70 | 2.58 ± 1.27 | 0.31 (0.14) | 2.27 | −0.90 (0.27) | −3.34 |
| PAQ-C-6 | 0.57 | | 0.48 | 0.61* | 0.70 | 2.48 ± 1.26 | 0.45 (0.14) | 3.30 | −0.74 (0.27) | −2.73 |
| PAQ-C-7 | 0.38 | | 0.54 | 0.63* | 0.70 | 2.81 ± 1.11 | 0.16 (0.14) | 1.15 | −0.62 (0.27) | −2.30 |
| PAQ-C-8 | 0.59 | | 0.52 | 0.61* | 0.70 | 2.47 ± 1.11 | 0.57 (0.14) | 4.22 | −0.22 (0.27) | −0.81 |
| PAQ-C-9 | 0.46 | | 0.57 | 0.52* | 0.70 | 3.06 ± 0.91 | 0.24 (0.14) | 1.76 | −0.31 (0.27) | −1.13 |
| PAQ-C score | | | | | | 2.68 ± 0.66 | 0.36 (0.14) | 2.70 | −0.10 (0.27) | −0.37 |

Notes:
* $p < 0.01$.
SD, Standard deviation; SE, Standard Error; EFA, Exploratory factor analysis based on principal axis factoring extraction with promax rotation (Kaiser Normalization), where rotation converged in three iterations.

### Confirmatory factor analysis

A 1-factor solution (Model-A) with all nine items of the original PAQ-C tool was found to be favored by the model fit indices, highest CFI, lowest RMSEA, a non-significant $\chi^2$ statistic, and lowest $\chi^2$/df. One of the items (PAQ-C-2) had a low loading value (*Comrey & Lee, 2013*); however, the model fit indices for the PAQ-C construct (Model-D) generated after deleting this item showed almost overlapping values (see Table 7 and Fig. 2).

### Item analysis and internal consistency

There was no substantial deviation from normality in the univariate distribution of the PAQ-C item scores, as evidenced by absolute values of skewness (<2), and absolute values

**Table 7 Fit statistics of the physical activity questionnaire-children (PAQ-C).**

| Models | CFI | GFI | RMR | RMSEA | $\chi^2$ | df | p | $\chi^2$/df |
|---|---|---|---|---|---|---|---|---|
| A | 0.950 | 0.937 | 0.073 | 0.059 (0.000–0.096) | 39.812 | 27 | 0.053 | 1.475 |
| B | 0.946 | 0.937 | 0.073 | 0.063 (0.013–0.100) | 39.811 | 26 | 0.041 | 1.531 |
| C | 0.946 | 0.937 | 0.073 | 0.062 (0.013–0.100) | 39.705 | 26 | 0.042 | 1.527 |
| D | 0.943 | 0.939 | 0.069 | 0.073 (0.028–0.114) | 34.555 | 20 | 0.023 | 1.728 |

**Note:**
A: 1-Factor, B: 2-Factor model (*Erdim, Ergün & Kuğuoğlu, 2019*; Turkish children), C: 2-Factor model (*Gobbi et al., 2016*; Italian children), D: 1-Factor model after deleting PAQ-C-2. CFI, Comparative Fit Index; GFI, Goodness of fit index; SRMR, Standardized root mean square residual; RMSEA, root mean square error of approximation.

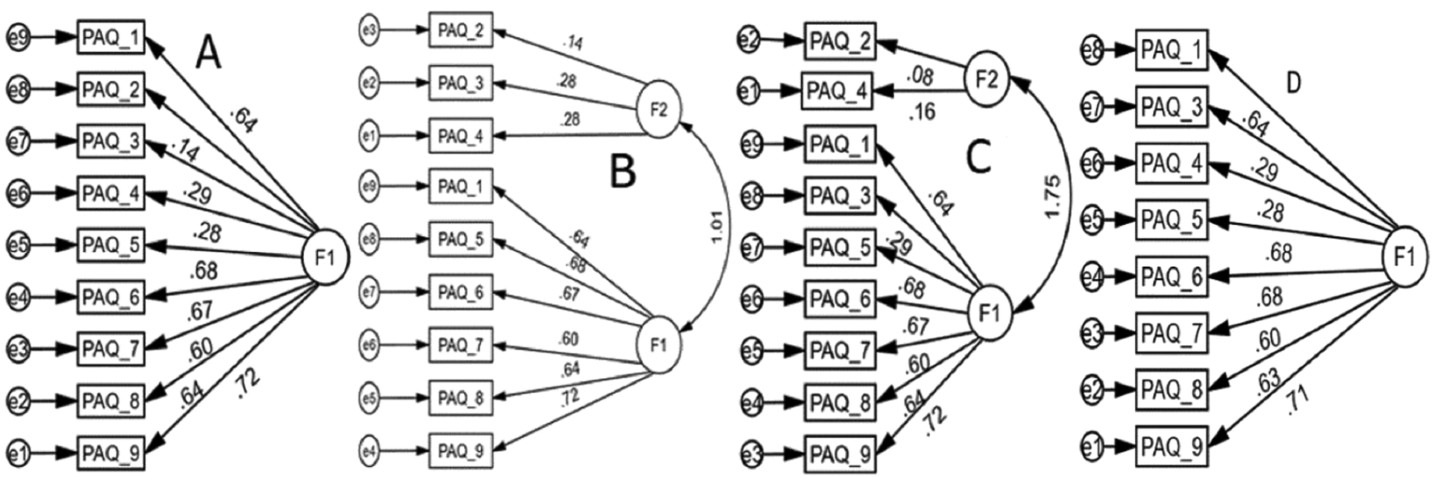

**Figure 2 Confirmatory factor analysis models of the physical activity questionnaire-children (PAQ-C) scores.** (A) 1-Factor, (B) 2-Factor model (*Erdim, Ergün & Kuğuoğlu, 2019*; Turkish children), (C) 2-Factor model (*Gobbi et al., 2016*; Italian children), (D) 1-Factor model after deleting PAQ-C-2. All coefficients are standardized. Ovals = latent variables, rectangles = measured variables, circles = error terms, single-headed arrows between ovals and rectangles = factor loadings, single-headed arrows between circles and rectangles = error terms.

of kurtosis (<7) (see Table 6) in the total sample (*Kim, 2013*). The values of item-total correlation, corrected item-total correlation, and Cronbach's alpha if an item was deleted, ranged from 0.20–0.57, 0.42–0.64, and 0.70–0.75, respectively (see Table 6). The PAQ-C showed a Cronbach's alpha of 0.74.

## DISCUSSION

To the best of our knowledge, this is the first study to investigate the psychometric validity of the PAQ-C in a sample of Arabic-speaking school-aged children of the Middle East and North Africa (MENA) region. In this study, statistical evidence showed that the PAQ-C (Arabic) has a unidimensional factor structure, adequate reliability implied by Cronbach's alpha, item-total correlation coefficients, and favorable item analysis, as implied by the measures of classical theory properties.

The factorial validity investigation was performed robustly after verifying that the dataset satisfied the necessary assumptions, and both EFA and CFA were performed on two randomly split sub-samples. In this study, the PAQ-C (Arabic) was found to have low average communalities of 0.48, and 0.50 in the two sub-samples (*MacCallum et al.,*

*1999*). The average factor loadings in the two sub-samples were 0.52, and 0.60, implying good to very good levels of correlations between measured variables, *i.e.*, item scores, and the factor score (*Comrey & Lee, 2013*). Moreover, a unidimensional factor structure of the PAQ-C (Arabic) was found valid, indicating a very high over-determination score (more number of items/indicators for each factor) (*MacCallum et al., 1999*). Therefore, combining all the pieces of evidence, and their interpretation regarding adequate sample size, sample sizes of 100 is adequate. Therefore, the sizes of the two sub-samples in this study, which comprised 137, and 190 Arabic-speaking school-aged children were adequate (*MacCallum et al., 1999*; *Manzar, Jahrami & Bahammam, 2021*). A direct comparison with most of the previous studies concerning values of measures that indicate the applicability of factorial validity was not possible because these studies did not report values of many of these measures. Three of the previous studies on the factor analysis of the PAQ-C did not report values of the diagonal element of the anti-image matrix, Bartlett's test of sphericity, determinant, KMO test of sampling adequacy, and communality (*Wang et al., 2016*; *Thomas & Upton, 2014*; *Gobbi et al., 2016*). A recent systematic review, and meta-analysis on the factorial validity of a widely used questionnaire tool, stressed the importance of verifying sample size adequacy using multiple measures (*Manzar, Jahrami & Bahammam, 2021*). *Erdim, Ergün & Kuğuoğlu (2019)* reported values of Bartlett's test of sphericity and KMO; these measures had similar values in our study. *Cuberek, Janíková & Dygrtfytfn (2021)* reported only KMO (0.8) values. The various sample adequacy measures give statistical evidence about the singularity, multicollinearity, correlations between items, shared variance, and correlations between items and factor scores. Therefore, in the absence of values of these measures, it is practically difficult to draw inferences from the factorial validity outcome of the previous studies (*Wang et al., 2016*; *Thomas & Upton, 2014*; *Gobbi et al., 2016*; *Manzar, Jahrami & Bahammam, 2021*; *Manzar et al., 2018*).

In accordance with the practical guidelines, in this study, multiple measures of factor retention including one robust measure (Monte Carlo parallel analysis) were employed (*Manzar, Jahrami & Bahammam, 2021*; *Manzar et al., 2018*; *Brown, 2015*). The inferences derived from EFA in previous studies may contain biases because of procedural issues such as the non-application of a robust measure of factor retention (*Wang et al., 2016*; *Thomas & Upton, 2014*; *Gobbi et al., 2016*; *Erdim, Ergün & Kuğuoğlu, 2019*; *Cuberek, Janíková & Dygrtfytfn, 2021*; *Manzar et al., 2018*; *Brown, 2015*). Three previous studies reported a two-factor structure based on two-factor retention criteria, namely, eigenvalue > 1 and cumulative variance rule in EFA (*Tomson et al., 2003*; *Gobbi et al., 2016*; *Erdim, Ergün & Kuğuoğlu, 2019*). Interestingly, in this study, a 2-factor structure was favored by two measures (eigenvalue > 1, and cumulative variance rule) of factor retention as well. However, the robust measure of the parallel analysis (Monte Carlo), and the Screen test indicated a 1-factor structure (*Manzar et al., 2018*; *Brown, 2015*). *Wang et al. (2016)* did not perform EFA, so they did not report values of factor retention measures used in EFA. Moreover, neither they did perform comparative CFA, nor did they justify the selection of a single factor model based on theoretical construct (*Wang et al., 2016*). In this study, most of the PAQ-C item scores satisfied the criteria for univariate

distribution, but a test of multivariate normality was not performed due to software limitations. Therefore, we used the principal axis factoring extraction method in EFA (*Costello & Osborne, 2005*).

A 1-factor model with all nine items as in the original PAQ-C was identified to be the best solution based on these considerations taken together: fit indices, non-improvement in fit indices after deletion of items showing lower factor loading(s), and IFC greater than one for two distinct 2-factor models (*Gobbi et al., 2016*; *Erdim, Ergün & Kuğuoğlu, 2019*). Fit indices of all models examined in the comparative CFA were nearly similar. Therefore, fit indices did not indicate which model has the best fit. To manage the problem of IFC greater than one for the two models with two factors, Bollen-Stine bootstrapping with 1,000 iterations was attempted to smoothen out multivariate non-normality (*Gobbi et al., 2016*; *Erdim, Ergün & Kuğuoğlu, 2019*; *Manzar et al., 2016*; *Bollen & Stine, 1992*). However, AMOS did not yield a solution but indicated an error message that the variances for one of the factors failed to be positive. Therefore, these 2-factor models were deemed inadmissible solutions (*Gobbi et al., 2016*; *Erdim, Ergün & Kuğuoğlu, 2019*). This left only two prospective solutions: a 1-factor model with all items and a 1-factor model after deleting items with low factor loadings. However, the model with deleted items did not have significantly better fit indices. Moreover, the PAQ-C item measuring activity during physical education classes, *i.e.*, the items with the low factor loading in this study showed optimum values for other measures, such as item-total correlations, corrected-item total correlations, and Cronbach's alpha if item deleted. Taking all these into account, *i.e.*, non-significant improvement in psychometric characteristics, along with the loss of information associated with the deletion of an item, a 1-factor solution with all the items of the original PAQ-C was found to be the most favored solution (*Albougami & Manzar, 2019*).

According to *Kim (2013)*, there are two separate sets of criteria for determining substantial deviations from the normality. As the sample size in this study was more than 300, therefore, either of these, absolute values of skewness (>2), or absolute values of kurtosis (>7) may imply substantial deviations from the normality (*Kim, 2013*). However, both, *i.e.*, all the PAQ-C items scores, and the total scores had absolute values of skewness less than 2, and absolute values of kurtosis less than 7 (*Nunnally, 1978*). The internal consistency, as determined by Cronbach's alpha, was satisfactory (*Nunnally, 1978*; *George & Mallery, 2003*). According to the widely used rule of thumb criteria by *George & Mallery (2003)*, a Cronbach's alpha value in the range of 0.7–0.8 is acceptable (*Nunnally, 1978*; *George & Mallery, 2003*). Previous studies have shown similar and slightly higher values of Cronbach's alpha in different populations. *Gobbi et al. (2016)* reported a Cronbach's alpha of 0.74 in Italian children. Slightly higher values were reported in Turkish (0.77) and Hong Kong children (0.79) (*Wang et al., 2016*; *Erdim, Ergün & Kuğuoğlu, 2019*). *Thomas & Upton (2014)* found higher values in English children, with values of 0.84 and 0.82 in two sub-samples. Evidence supported an adequate item discrimination ability for all the item scores of the PAQ-C in this study population based on the classical theory parameters. This was implied by the fact that all values of item-total correlation

coefficients, corrected-item total correlation coefficients, and Cronbach's alpha if an item was deleted, were more than 0.2 (*Wang et al., 2017*).

Some of the important limitations that might have a bearing on the generality of the inferences are mentioned here. Firstly, a lack of concurrent validation concerning an objective measurement tool such as an accelerometer may limit inference deductions about the PAQ-C's accuracy. Secondly, a test-re-test was not performed since schools were closed due to COVID-19 lockdown; therefore, this may limit temporal reliability interpretations. Future studies using robust measures of item response theory may help in further establishing item-level psychometric parameters of the PAQ-C.

## CONCLUSION

A 1-factor model of the PAQ-C (Arabic) with all original nine items had adequate reliability, internal homogeneity, and favorable item analysis. The original English PAQ-C was developed for the Canadian population living in colder conditions. Despite this, its Arabic version with cultural adaptation was found to be valid among a sample of school-aged children in Saudi Arabia. The findings of this study affirm that the PAQ-C (Arabic) is a suitable tool to determine moderate to vigorous physical activity in children in Arabic countries. Future research addressing the psychometric properties of the PAQ-C in repeated administrations, different seasons, and multi-centric data collection strategies is warranted.

### Funding
This work was supported by the Deanship of Scientific Research at Majmaah University (project no. 1439/88). The funders had no role in study design, data collection and analysis, decision to publish, or preparation of the manuscript.

### Grant Disclosures
The following grant information was disclosed by the authors:
Deanship of Scientific Research at Majmaah University: 1439/88.

### Competing Interests
The authors declare that they have no competing interests.

### Author Contributions
- Mohamed Sherif Sirajudeen conceived and designed the experiments, performed the experiments, analyzed the data, prepared figures and/or tables, authored or reviewed drafts of the paper, and approved the final draft.
- Mohamed Waly conceived and designed the experiments, performed the experiments, analyzed the data, prepared figures and/or tables, authored or reviewed drafts of the paper, and approved the final draft.

- Md. Dilshad Manzar conceived and designed the experiments, analyzed the data, prepared figures and/or tables, authored or reviewed drafts of the paper, and approved the final draft.
- Mazen Alqahtani conceived and designed the experiments, authored or reviewed drafts of the paper, and approved the final draft.
- Msaad Alzhrani conceived and designed the experiments, authored or reviewed drafts of the paper, and approved the final draft.
- Ahmad Alanazi conceived and designed the experiments, authored or reviewed drafts of the paper, and approved the final draft.
- Radhakrishnan Unnikrishnan performed the experiments, prepared figures and/or tables, and approved the final draft.
- Hariraja Muthusamy performed the experiments, prepared figures and/or tables, and approved the final draft.
- Rashmi Saibannavar performed the experiments, prepared figures and/or tables, and approved the final draft.
- Wafa Alrubaia performed the experiments, prepared figures and/or tables, and approved the final draft.

### Human Ethics

The following information was supplied relating to ethical approvals (*i.e.*, approving body and any reference numbers):

The Majmaah University Ethical Committee approved this research (no. MUREC-Feb.19/COM-2020/21-5).

### Data Availability

The raw data is available in the Supplemental Files.

### Supplemental Information

Supplemental information for this article can be found online at http://dx.doi.org/10.7717/peerj.13237#supplemental-information.

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
