# Peer review of "Physical activity questionnaire for older children (PAQ-C): Arabic translation, cross-cultural adaptation, and psychometric validation in school-aged children in Saudi Arabia"

_PeerJ, doi:10.7717/peerj.13237_

## Round 0.1 · original submission · Major Revisions

The article has merit but some changes must be done aiming to improve the reproducibility.

Reviewer 1 ·

Basic reporting

• The English language could be clearer and more fluid;
• The references used are specific and well presented although they are mostly a bit old.
• The structure seems to be conform the standards PeerJ;
• Figures and tables are relevants, well labeled and described, however two of them deserve correction and improvement.
• The necessary data has been provided.

Experimental design

• Seeks to harness an existing questionnaire in another language for use in another paísto.
• Was it not clear how the 8-year-olds answered the questionnaire? It was the same way as the 14! Their understanding was guaranteed?
• The procedures (line 110-113) should clearly state how the data on personal characteristics were collected, the tool used and the way in which they were applied.
• It was not clear whether the translators who participated in the study were certified official translators!
• What are the characteristics of the group of experts who participated in the study, can we consider them specialists!? Its general characteristics that guarantee its competence must be described (Paixão, Abad, & Giménez, 2018).

Validity of the findings

• The fact that the collection tool is based on a tool that has already been validated in other countries, guarantees more robust scientific knowledge.
• Its application in different contexts is important for the conclusions to be strengthened.
• In which the author and the reference value is based to affirm that the psychometric validity is adequate!
• Why choose a scale of 5? It must be backed up with references, and if 1 is “no activity” and 5 is “seven times or more” what do 2, 3 and 4 mean? The options must be closed and clearly know how many times a week it corresponds.
• The reference that justifies the validity and significance is very old (1999), a more current reference should be used (Brewer & Jones, 2002; Butragueño & Benito, 2014; Simón, Fernández & Contreras, 2017; Saldaña & García, 2013; Otero, González & Calvo, 2012).
• It seems to me that the conclusion turns out to be more of a result than the conclusions that can be drawn from the study. Information is also presented in a very telegraphic and unreflective way.
• Line 215: It seems to me that the term “empirical” evidence is not suitable for statistical tests, it should be revised.
• In my view the statistical procedures expressed in the results demonstrate statistical robustness.
• An effort should be made to review the Discussion to improve understanding and readability, clarifying the interpretation of the results.
• Line 258: when it is stated that the internal consistency is satisfactory, the value must be presented.
• Line 270: must be presented a reason why the test-retest was not done! Since it is reported in the introduction to be carried out by other studies.

Additional comments

Introduction:
As it is a validation article, it lacks further grounding in this part according to the validation tools used in the process, as the introduction is very much directed only to physical activity.
References:
Reference 4 on line 286 does not appear in the text, however the references presented may be more current, as in this case the theme has been developed in other studies after 1998;
Images and tables:
Table 1: it should have only horizontal lines as the other tables presented;
Figure 1: must also have the legend of the other two lines of data (green and red) and be presented without color, in gray scale;

Reviewer 2 ·

Basic reporting

It should be emphasized why the study is important for this sample (Arabic). It should also be stated why the development of the scale is important.Why is it important to develop an Arabic version of this scale?

Introduction section, Information on physical inactivity (maybe numbers for inactivity), especially children in Arabia should be given.

In introduction, it should be explained which theory the study (especially phsyical activity scale) is related to. The phsyical activity should be based on the theory.

Current literature should be used in the introduction. It should be written using current literature (2021, 2020) in a specific flow order as a link between topics.

The hypotheses of the study should be stated.

The English language of the text should be improved

Experimental design

In methods section, cluster sampling method should be based on the scientific sources. A detailed explanation should be given about the sampling method. For example; How many schools have been identified? How many of these were sampled?
How the sample size was calculated should be stated based on scientific sources.
An explanation should be given about where, how and by whom the data is collected.
More detailed information should be given about the characteristics of the participants.
Authors should be informed about the criteria for inclusion and exclusion of participants from the study.

Validity of the findings

More detailed information should be given in the conclusion section. Also, the recommendations section should be added.
Discussion section should be developed.More references should used.

Additional comments

References should be arranged according to the journal format.
According to the quality of this journal, the study is quite simple in terms of methods section.
The aim of the development of the arabic version of the scale or the innovations which the scale will bring to the sports field have not been fully emphasized.

---

## Round 0.2 · accepted · Accept

The article was improved and the current version is ready for publication.

Reviewer 1 ·

Basic reporting

Overall, the important issues raised in the previous review have been clarified.

Experimental design

The authors satisfactorily met all the needs identified in the previous review.

Validity of the findings

All the changes resulting from the review contributed to a clear improvement of the document, increasing its quality in terms of scientific production.

Additional comments

After the reformulation carried out by the authors, taking into account the different notes of the reviewers, I understand that the article is now ready to be published.

Reviewer 2 ·

Basic reporting

The authors made major changes to each part of the article. The article is ready for publication

Experimental design

The authors made major changes to each part of the article. The article is ready for publication

Validity of the findings

The authors made major changes to each part of the article. The article is ready for publication